# Peer review of "Navigating the Dynamic Landscape of SARS-CoV-2: The Dual Role of Neutralizing Antibodies, Variability in Responses, and Strategies for Adaptive Pandemic Control"

_covid, doi:10.3390/covid4090100_

Round 1
Reviewer 1 Report
This is a well-written review, summarizing essential knowledge of the neutralizaing antibodies. I like the clear model of different ages very much. The model is clear and expressive. Together with the easy-to-follow writing, the information is simple to be understood.
The authors stated in the title "Strategies for Adaptive Pandemic Control", but this part was insufficiently written. This part is particularly interesting and thus worth more summarization.
Figure 1 seems not necessary. Such a stratification is useful in a meta-analysis, but not in a review.
Author Response
|
Reviewer 1 |
|
|
Comments |
Response |
|
The authors stated in the title "Strategies for Adaptive Pandemic Control", but this part was insufficiently written. This part is particularly interesting and thus worth more summarization. |
Thank you for your suggestions. Changes have been done and are also highlighted. |
|
Figure 1 seems not necessary. Such a stratification is useful in a meta-analysis, but not in a review. |
Changes to the manuscript has been done accordingly. |
Reviewer 2 Report
No comments
Minor comments
.- Improve presentation of tables (more space between words)
.- Cite references appropriately in order of mention ( from line 301 )
.- unify references style
.- In the text, expand the information about the table 4
.- Figure 4. Adequately indicate what it represents
Author Response
|
Reviewer 2 |
|
|
Comments |
Response |
|
Improve presentation of tables (more space between words) Cite references appropriately in order of mention ( from line 301 ) unify references style. In the text, expand the information about the table 4 Figure 4. Adequately indicate what it represents
|
Thank you for your suggestions. All the changes suggested by you are done in the manuscript and highlighted. |
Reviewer 3 Report
Please see the attached document for this.
Please see the attached document for this.

Author Response
|
Reviewer 3 |
|
|
Comments |
Response |
|
Some introductions of challenges in targeted therapeutic interventions and vaccination approaches on SARS-COV-2 should be included as listed in the abstract section. |
Thank you for your suggestions. Changes in the manuscript have been done accordingly. |
|
2. In figure one, I just wonder how would the authors screen their records, what methodology was used for this? Can the authors describe that in the manuscript? |
Changes in the manuscript have been done as per the suggestions received. |
|
3. Line 163 – 172, they authors described the major functions of neutralization antibody in only one sentence. They authors can spread this into multiple sentences or simply using number notations to make this part clear. |
Changes in the manuscript have been done accordingly. |
|
4. Line 167, preventing there attachment. “There” typo. |
Changes in the manuscript have been done accordingly. |
|
5. Table 1, sub-title “Name of Neutralization Agents”, I think: “Type of Neutralization Agents” fits this section. 6. Line 234, after citation #30, should be “.” Not |
Changes in the manuscript have been done as per the suggestions received. |
|
7. Same to Line 238 after citation #31. |
Changes in the manuscript have been done accordingly. |
Round 2
Reviewer 1 Report
All issues addressed.
The authors added the section "Strategies for Adaptive Pandemic Control". Good job!